# Comparative evaluation of iStent versus iStent inject W combined with phacoemulsification in open angle glaucoma

Shuu Morita *, Yoshihito Sakanishi , Ikari Riyu, Satoshi Watanabe, Nobuyuki Ebihara

Department of Ophthalmology, Juntendo University Urayasu Hospital, Tiba, Japan

* s-morita@juntendo.ac.jp

## Abstract

### Purpose

The study aimed to compare the 12-month post-operative outcomes of iStent and iStent inject W (inject W), and the factors associated with their success in open-angle glaucoma.

### Methods

This single-center, retrospective comparative case series evaluated the medical records of patients who underwent iStent (comprising 1 stent) or inject W (comprising 2 stents) implantation with cataract surgery for primary open-angle glaucoma and normal tension glaucoma between January 2019 and March 2022. The 12-month post-operative efficacy outcomes included intraocular pressure (IOP), glaucoma medications, and survival analysis of the probability of success. "Failure" was defined as any of the following conditions compared to baseline: 1) IOP elevation, 2) increased glaucoma medication, or 3) IOP decline not exceeding 20% when glaucoma medication scores were comparable, and 4) need for additional glaucoma surgery. The safety outcomes included intra- and post-operative adverse events and changes in the best-corrected visual acuity and visual field.

### Results

The study comprised 55 eyes in the iStent and 105 in the inject W groups. At 12 months, treatment success was achieved in 66.0% of iStent and 78.4% of inject W eyes. The mean IOP was lower, and the percent reduction from baseline was equal in iStent-treated eyes (8.0% reduction, 14.8 mmHg to 13.7 mmHg, $P<0.01$) and inject W-treated eyes (11.9% reduction, 15.0 mmHg to 13.8 mmHg, $P<0.01$) (between-group comparison, $P = 0.23$). The mean medication burden decreased significantly from 2.5 to 1.1 for iStent (55.0% reduction, $P<0.01$) and 2.9 to 1.7 for iStent inject (46.8% reduction, $P<0.01$), with no significant differences between the two groups ($P = 0.17$). Both devices exhibited excellent safety.

**Data Availability Statement:** All relevant data are within the manuscript and its Supporting information files.

**Funding:** The author(s) received no specific funding for this work.

**Competing interests:** The authors have declared that no competing interests exist.

## Conclusions

Both devices significantly reduced IOP and glaucoma medication 12 months post-operatively. The outcome measures did not differ significantly between the two groups, and lower baseline IOP was predictive of surgical failure.

## Introduction

Blindness is a prevalent sequela of glaucoma worldwide; glaucoma is the leading cause of blindness in Japan [1–3]. Intraocular pressure (IOP) is a risk factor for the development and progression of open-angle glaucoma (OAG), and IOP reduction is the only intervention proven to retard disease progression [4]. Glaucoma medications, laser therapy, and surgery have been used to reduce IOP. Currently, topical glaucoma medications with multiple mechanisms of action are available, but multidrug treatment is associated with problems such as poor compliance and adverse effects [5–8]. Moreover, conventional filtration surgery and glaucoma drainage devices reportedly cause serious complications, such as hypotony and suprachoroidal hemorrhage [9–11]. The recently developed minimally invasive glaucoma surgeries (MIGS) are expected to achieve IOP reduction more safely. The Trabecular Micro-Bypass Stent Systems, iStent (iStent) and iStent inject W (inject W) (Glaukos Corporation, Laguna Hills, CA, USA), are available in Japan for MIGS.

Both iStent (first-generation) and iStent inject W (second-generation), which is an improved version of iStent inject (inject), aim to reduce IOP by improving conventional trabecular outflow through Schlemm's canal (SC), bypassing the trabecular meshwork (TM) that has the highest resistance to aqueous outflow. The aqueous humor flows into SC through the stent's lumen and drains through a normal physiological outflow pass-way, including collector channels (CCs) and aqueous veins. The iStent is mounted on a dedicated injector. The half-cut pipe is inserted oblique to the TM so that this section faces toward the outside of SC and struck horizontally to bypass the TM to reach the SC, thereby lowering the IOP. The iStent and inject W are available in Japan. The inject W is equipped with two dedicated injectors, both of which can be inserted into one eye. The device bypasses the TM to reach the SC by driving perpendicularly into the TM.

Many studies have reported the benefit and safety of iStent and iStent inject. A prospective randomized controlled trial reported that iStent with phacoemulsification offered superior IOP and glaucoma medication reduction compared to phacoemulsification alone [12]. Moreover, no serious complications, such as suprachoroidal hemorrhage, anterior chamber loss, or endophthalmitis [12, 13], have been reported with iStent, which is considered a highly safe device. A prospective, randomized, comparative study of iStent inject with phacoemulsification vs phacoemulsification alone found that the IOP was reduced by 7.0 mmHg with iStent inject with phacoemulsification compared to a reduction of 5.4 mmHg with phacoemulsification alone [14]. The rate of IOP reduction exceeding 20% without glaucoma medication was also better in the group treated with iStent inject, and there was no difference in complications. However, there are no study has investigated iStent inject W, a modified version of inject.

In the present study, we compared the 12-month outcomes of iStent and inject W in OAG limited to primary open-angle glaucoma (POAG) and normal tension glaucoma (NTG). We also investigated the factors associated with the success of iStent and iStent inject W.

## Materials and methods

### Study design

The single-center, retrospective comparative case series study reviewed the medical records of 160 eyes of 114 patients who underwent phacoemulsification combined with iStent or iStent inject W implantation for OAG (POAG and NTG) at Juntendo University Urayasu Hospital between January 2019 and March 2022. All surgeries were performed at a single center by five surgeons. The choice of implant was based primarily on availability in the Japanese market. Only the first-generation trabecular micro-bypass (iStent) was available for use from 2019 to 2021. After the receiving approval in 2021, the second-generation implant (inject W) was used for IOP reduction in most cases. The study was conducted according to the tenets of the Declaration of Helsinki; ethical approval was obtained from the institutional review board of Juntendo University (E21-0058-U01). The surgeon explained the purpose of the study to the patients, and consent was documented in their medical records.

### Inclusion and exclusion criteria

Patients with 1) a decrease in visual acuity (VA) due to cataract, 2) IOP controlled below 25 mmHg with topical glaucoma medication, 3) no rapid progression of visual field (VF) defects, and 4) POAG or NTG were included in this study. The exclusion criteria were patients under oral carbonic anhydrase inhibitor therapy, those who had undergone laser therapy or other surgical treatments for glaucoma, corneal transplantation, history of vitreoretinal disease (age-related macular degeneration, diabetic retinopathy, or retinal vein occlusion), and history of vitrectomy. In addition, patients in whom the iStent or iStent inject W was attempted but unsuccessfully inserted were excluded from the study.

Baseline demographic data, including age, sex, glaucoma type, surgeon, IOP, and the score of glaucoma medications (single: 1, combination: 2), best-corrected visual acuity (BCVA), and VF were collected. IOP was measured using Goldmann applanation, and BCVA was measured using decimal VA. BCVA was calculated by converting decimal VA to the logarithm of the minimum angle of resolution (logMAR), and the VF was evaluated by the mean deviation value obtained from the Humphrey Field Analyzer (HFA, ZEISS, Oberkochen, Germany). Post-operatively, IOP and the glaucoma medication score were recorded on day 1, week 1, and 1, 3, 9, and 12 months. If the IOP was greater than the pre-operative IOP at any follow-up visit, the patient could resume one glaucoma medication per visit. Intraoperative/post-operative complications and secondary surgeries were also investigated. The post-operative BCVA was compared to the corresponding pre-operative values on day 1, week 1, months 1 and 3, and at the final visit after surgery. The post-operative VF was compared to the pre-operative VF.

### Surgical procedure (device and surgical technique)

The iStent and inject W trabecular micro bypass devices are designed to create one or two patent bypasses. When implanted through the TM into SC, the stents are designed to increase the outflow of aqueous humor from the anterior chamber, thereby decreasing the IOP. Device implantation was performed after phacoemulsification. The self-trephining tip of the iStent was used to penetrate the TM through a temporal clear corneal incision and under direct-gonioscopic visualization. Once the device was placed in the SC, the inserter button was depressed to release the device. The inserter tip was used to fully drive the iStent into the canal. For iStent inject W implantation, the sleeve of the injector was retracted, revealing the trocar and micro insertion tube. The trabecular meshwork was penetrated by the trocar, and the delivery button was depressed to implant the first stent, and the second stent was placed 1–3

clock hours later. Following surgery, patients were administered a topical antibiotic (1–2 weeks) and topical anti-inflammatory agent, and the dose was tapered over 4–12 weeks.

### Outcome measures

The primary outcome measures were within-group and between-group comparisons of IOP and the score of the glaucoma medications. The efficacy outcomes included mean IOP reduction, reduction in the mean score of glaucoma medications, and survival analysis of the success probability. "Failure" was defined as any of the following events compared to baseline: 1) increase in IOP, 2) increase in the glaucoma medication score, or 3) IOP did not decrease by more than 20% when the glaucoma medication scores were comparable, and 4) need for additional glaucoma surgery. The safety outcomes included intra- and post-operative adverse events, secondary surgeries, and changes in BCVA and VF.

### Statistical analysis

The chi-squared and Mann–Whitney U tests were used to analyze the categorical and continuous variables, respectively. Continuous variables were presented as the mean ± standard deviation. Proportions (%) were used to describe categorical variables. For data analysis, decimal VA measurements were converted to the logMAR equivalent. IOP changes were evaluated using Friedman's test for multiple comparisons, followed by the Wilcoxon signed-rank test in each group, while the Holm method was used for $P$-value adjustment. The Mann–Whitney U test was used to evaluate whether or not there was a significant difference in IOP change and IOP reduction rate between the two groups. Changes in glaucoma medication in each group were also evaluated via the same methods. Kaplan–Meier survival analysis and log-rank tests was used to report the cumulative surgical success rate. Cox proportional hazard models was employed to identify factors predictive of surgical failure. The predictors of failure were analyzed using univariate and multivariate regression analyses. Variables with $P$ values $\leq 0.1$ in the univariate analysis were entered into the multivariate analysis.

$P$ values $<0.05$ were considered statistically significant. All statistical analyses were performed using R version 4.1.2 software for Windows (R Foundation for Statistical Computing, Vienna, Austria).

## Results

### Baseline characteristics

The patients' demographic characteristics are outlined in Table 1. This study investigated 55 eyes of 40 patients (mean age: 71.5±9.5 years) in the iStent group and 105 eyes of 74 patients (mean age: 74.3 ± 7.8 years) in the inject W group. Patients in the iStent group ($P$ = 0.04) were younger, but there was no difference in sex ratio or disease type. The pre-operative BCVA was similar, but the VF was worse in the inject W group ($P<0.01$). There were no differences in the pre-operative IOP and glaucoma medication scores.

### Intraocular pressure/glaucoma medication score

The mean IOP decreased significantly from 14.8 ± 2.2 mmHg pre-operatively to 13.7 ± 2.9 mmHg 12 months post-operatively in the iStent-treated eyes ($P$ = 0.02). The mean IOP also decreased significantly from 15.0 ± 2.8 mmHg pre-operatively to 13.8±3.3 mmHg 12 months post-operatively in the inject W-treated eyes ($P < 0.01$) (Table 2).

The medication scores ranged from 2.5±1.3 pre-operatively to 1.1±1.4 at 12 months post-operatively in the iStent eyes ($P<0.01$) and from 2.9±1.4 pre-operatively to 1.7±1.6 at 12

**Table 1. Patient demographics.**

|  | iStent | inject W | p value |
|---|---|---|---|
| n (eyes) | 55 | 105 |  |
| Age (mean ±SD, range) | 71.5±9.5 | 74.3 ±7.8 | **0.04** |
| Operated eye: right [n, (%)] | 24 (43.6) | 49 (46.7) | 0.74 |
| Sex: male [n, (%)] | 27 (49.1) | 66 (62.9) | 0.13 |
| Glaucoma type: POAG [n, (%)] | 40 (72.7) | 64 (61.0) | 0.16 |
| Pre-operative BCVA (mean ±SD, range) | 0.33±0.40 | 0.38±0.43 | 0.42 |
| Pre-operative VF (dB) (mean ±SD, range) | -7.4±4.7 | -11.1±7.4 | **<0.01** |
| Pre-operative IOP (mmHg) (mean ±SD, range) | 14.8±2.2 | 15.0±2.8 | 0.85 |
| Pre-operative medication score (mean ±SD, range) | 2.6±1.3 | 2.9±1.4 | 0.11 |

SD: Standard Deviation; POAG: Primary Open Angle Glaucoma; BCVA: Best-Corrected Visual Acuity (the decimal visual acuity was converted to logMAR and described); VF: Visual Field, the mean deviation in the *Humphrey* Field Analyzer; IOP: Intraocular Pressure

Categorical variables: chi-squared test

Continuous variables: Mann–Whitney U test

Age was younger in the iStent-treated eyes, and the VF was worse in the inject-treated W eyes.

months post-operatively in the inject W eyes ($P<0.01$) (Table 2). We examined whether the two groups differed with respect to the change in IOP and glaucoma medication score at 12 months post-operatively compared to baseline, but none of the differences were significant ($P = 0.23$, $P = 0.17$) (Table 3).

## Cumulative survival rate

The Kaplan–Meier survival curve rates at 6 and 12 months post-operatively in the iStent eyes were 83.4% and 66.0%, respectively, while the corresponding values in the inject W eyes were 84.0% and 78.4%, respectively (Fig 1). There was no statistically significant difference in the survival-curve rates between the iStent eyes and inject W eyes ($P = 0.43$). The survival-curve

**Table 2. IOP and glaucoma medication score: iStent eyes and inject W eyes.**

|  |  | iStent eyes | | | | inject W eyes | | | |
|---|---|---|---|---|---|---|---|---|---|
|  |  | IOP (mmHg) | | Medication Score | | IOP (mmHg) | | Medication Score | |
| Pre-operative |  | 14.8±2.2 | p value | 2.5±1.3 | p value | 15.0±2.8 | p value | 2.9±1.4 | p value |
| Post-operative | 1day | 14.3±3.6 | 1.00 |  |  | 13.4±5.0 | 0.19 |  |  |
|  | 1W | 16.0±4.6 | 0.92 | 0.2±0.9 | **<0.01** | 16.1±4.8 | 1.00 | 0.2±0.8 | **<0.01** |
|  | 1M | 14.4±3.2 | 1.00 | 0.7±1.2 | **<0.01** | 13.6±2.9 | **<0.01** | 1.0±1.4 | **<0.01** |
|  | 3M | 13.9±2.9 | **0.02** | 0.8±1.3 | **<0.01** | 13.4±2.7 | **<0.01** | 1.5±1.5 | **<0.01** |
|  | 6M | 13.3±2.5 | **<0.01** | 0.9±1.3 | **<0.01** | 13.8±2.7 | 0.05 | 1.7±1.5 | **<0.01** |
|  | 12M | 13.7±2.9 | **0.02** | 1.1±1.4 | **<0.01** | 13.8±3.3 | **<0.01** | 1.7±1.6 | **<0.01** |

Each group was compared pre-operatively and at each follow-up point.

IOP: Intraocular Pressure, W: Week, M: Months

Friedman test: p value<0.01

Wilcoxon signed-rank test

*P*-value adjustment: Holm method

**Table 3. Comparison of changes in IOP and medication score at 12 months postoperatively: iStent eyes vs inject eyes.**

| | IOP (mmHg) | | | Glaucoma medication score | | |
|---|---|---|---|---|---|---|
| | iStent | inject W | p value | iStent | inject W | p value |
| Pre-operative | 14.8±2.2 | 15.0±2.8 | | 2.5±1.3 | 2.9±1.4 | |
| POM 12 | 13.7±2.9 | 13.8±3.3 | | 1.1±1.4 | 1.7±1.6 | |
| Change | -1.2±2.2 | -1.9±2.3 | 0.23 | -1.4±1.3 | -1.2±1.2 | 0.17 |
| | (-8.0±15.4%) | (-11.9±14.7%) | | (-55.0±63.3%) | (-46.8±43.8%) | |

IOP: Intraocular Pressure; POM: Postoperative month

Mann–Whitney U test

No significant difference in the changes in IOP and glaucoma medication score at 12 months postoperatively between the iStent eyes and inject W eyes

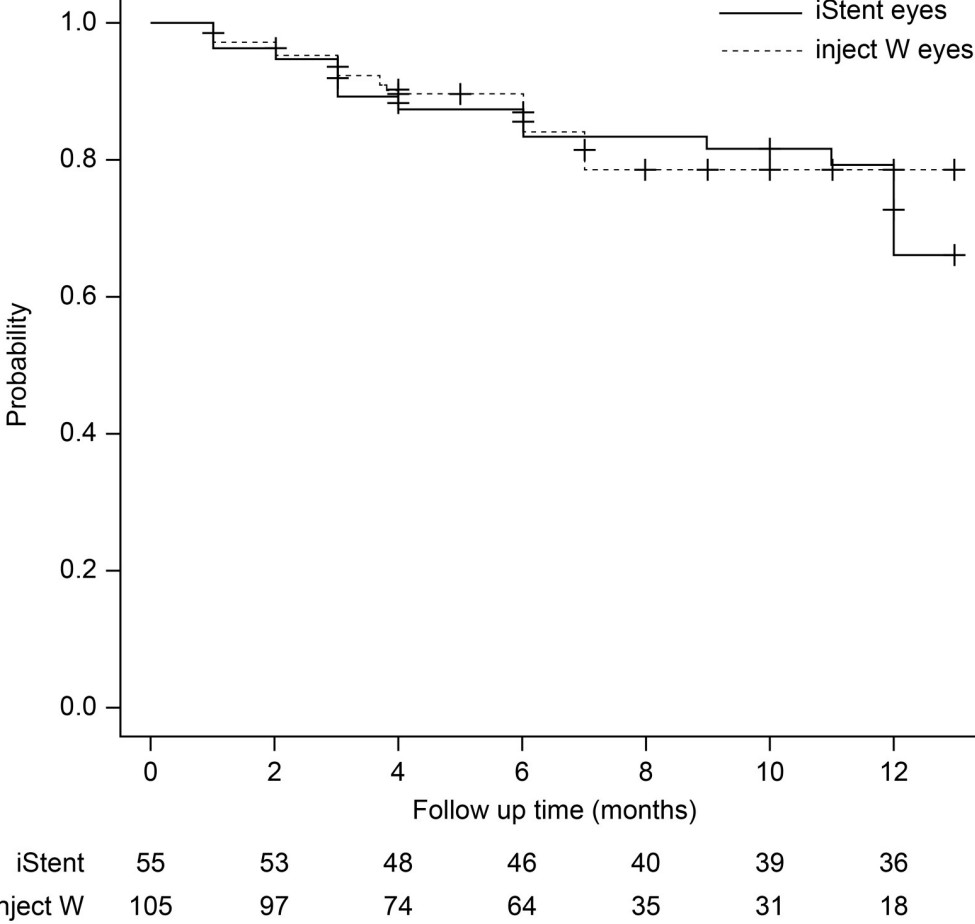

**Fig 1. Comparison of cumulative survival rates: iStent eyes vs. inject W eyes.** The cumulative survival rates at 6 and 12 months postoperatively for the iStent eyes were 83.6% and 66.0% and 84.0% and 78.4% for the inject W eyes, respectively. Log-rank test: p = 0.50. Cox proportional hazard model: HR, 0.74; 95%CI, 0.35–1.56; p = 0.43. HR: Hazard ratio, CI: Confidence interval.

rates corrected by patient background using the Cox proportional hazard model also showed similar results (hazard ratio: 0.74, 95% confidence interval: 0.35–1.56, $P$ = 0.43).

## BCVA and VF

In the iStent-treated eyes, the pre-operative BCVA was 0.33±0.40 logMAR, which improved to 0.13±0.30 logMAR on post-operative day 1 ($P$<0.01) (Table 4). The BCVA was maintained, with a final VA of 0.05±0.26 logMAR. The VF was -7.4±4.7 dB pre-operatively versus -8.0±4.7 dB post-operatively, with no significant difference ($P$ = 0.75) (Table 5). The inject W eyes showed improvements in the BCVA from 0.38±0.43 logMAR pre-operatively to 0.12±0.34 logMAR on day 1 post-operatively ($P$<0.01), which was maintained with a final BCVA of 0.08 ±0.33 logMAR (Table 4). The VF was -11.1±7.4 dB pre-operatively and -10.4±8.5 dB post-operatively ($P$<0.01) (Table 5).

## Safety

Table 6 shows the complications recorded during this study. No serious complications, such as suprachoroidal hemorrhage or endophthalmitis, were observed. The complications did not differ significantly between the two groups. Both iStent and inject W eyes experienced anterior chamber hemorrhage and transient IOP elevation, but additional surgery was not required. Transient IOP elevation also improved upon follow-up observation or resumption of topical glaucoma medication.

## Factors affecting cumulative survival

In the iStent eyes, the univariate analysis showed that disease type, pre-operative IOP, surgeon, and anterior chamber hemorrhage were significant ($P\leq$ 0.1) factors, while according to the multivariate analysis, only pre-operative IOP was a significant factor (Table 7). In the inject W group, pre-operative IOP and surgeon were significant factors in the univariate analysis with $P\leq$ 0.1, and pre-operative IOP was the only significant factor in the multivariate analysis (Table 7). In both groups, a higher pre-operative IOP was associated with a lower likelihood of failure.

## Discussion

Several retrospective have compared iStent with iStent inject. However, our study is valuable owing to the scarcity of studies comparing the outcomes of iStent and inject W [15]. In addition, few studies have reported on the follow-up results of Inject W beyond one year [15]. The

**Table 4. The transition of BCVA: iStent eyes and inject W eyes.**

| | | iStent | | inject W | |
|---|---|---|---|---|---|
| | | BCVA | p value | BCVA | p value |
| Pre-operative | | 0.33±0.40 | | 0.38±0.43 | |
| Post-operative | 1 day | 0.13±0.30 | **<0.01** | 0.12±0.34 | **<0.01** |
| | 1 week | 0.04±0.13 | **<0.01** | 0.04±0.18 | **<0.01** |
| | 1 month | 0.06±0.26 | **<0.01** | 0.08±0.28 | **<0.01** |
| | 3 months | 0.06±0.26 | **<0.01** | 0.12±0.36 | **<0.01** |
| | Last-visit | 0.05±0.26 | **<0.01** | 0.08±0.33 | **<0.01** |

BCVA: Best-Corrected Visual Acuity, decimal visual acuity was converted to the logMAR and described. Friedman test: p<0.01. Wilcoxon signed-rank test. *P*-value adjustment: Holm method. Both groups showed improvement in BCVA from the day after surgery.

**Table 5. The transition of VF: iStent eyes and inject W eyes.**

| | iStent | | inject W | |
|---|---|---|---|---|
| | VF (dB) | p value | VF (dB) | p value |
| Pre-operative | -7.4±4.7 | | -11.1±7.4 | |
| Post-operative | -8.0±4.7 | 0.746 | -10.4±8.5 | **<0.01** |

VF: Visual Field, mean deviation in the *Humphrey* Field Analyzer

Wilcoxon signed-rank test

In both groups, no worsening of VF was observed postoperatively compared to baseline.

study also examined the safety and factors associated with the success or failure of Inject W. Our study reported some of the first comparative 12-month outcomes of the first- and second-generation trabecular micro-bypass stents, viz. iStent and inject W, with phacoemulsification for POAG and NTG. Both devices yielded significant reductions in IOP and medications while maintaining excellent safety. Moreover, a higher pre-operative IOP was associated with less failure.

## Efficacy

At 12 months post-operatively, IOP reduction was 1.2 ± 2.2 mmHg (8.0 ± 15.4%) in the iStent eyes and 1.9 ± 2.3 mmHg (11.9 ± 14.7%) in the inject W eyes, with no difference between the two groups ($P$ = 0.23). Every 1 mmHg of IOP reduction is correlated with a 10% lower risk of glaucoma development [16] and a 10–19% lower risk of glaucoma progression [17, 18]. Thus, IOP reductions achieved with minimally invasive methods, such as iStent and inject W, are clinically substantial.

No statistically significant difference was observed between the two devices in terms of the glaucoma medication score, with iStent and inject W eyes showing a decrease of 1.4±1.3 and 1.2±1.2 agents, respectively The medication reduction benefits of these devices are highly variable and immensely impactful. Multidrug treatment degrades the conjunctiva and increases the risk of future surgical failure [19]. Reduced compliance has also been reported to limit the efficacy of glaucoma medication and increase the risk of progressive VF defects [20]. It is also

**Table 6. Complications.**

| | iStent | inject W | p value |
|---|---|---|---|
| | n = 55 | n = 105 | |
| | n (%) | | |
| Coagula in the anterior chamber | 1 (1.8) | 9 (8.6) | 0.17 |
| Hyphema | 0 | 1 (1.0) | 1.00 |
| Transient IOP elevation: 10 mmHg above baseline | 3 (5.5) | 6 (5.7) | 1.00 |
| Others (iritis, macular edema, etc.) | 0 | 6 (5.7) | 0.10 |
| Total number of complications | 4 (7.3) | 17 (16.2) | 0.14 |

IOP: Intraocular Pressure

Chi-squared test

There were no serious complications, such as endophthalmitis or suprachoroidal hemorrhage, and no complications requiring additional surgery.

Hyphema improved within 2 weeks of follow-up.

Transient IOP elevation improved with follow-up or resumption of glaucoma medication.

**Table 7. Factors affecting cumulative survival: iStent eyes and inject W eyes.**

| | iStent eyes | | | | | | | | inject W eyes | | | | | | | |
|---|---|---|---|---|---|---|---|---|---|---|---|---|---|---|---|---|
| | the univariate analysis | | | | the multivariate analysis | | | | the univariate analysis | | | | the multivariate analysis | | | |
| | HR | 95% CI | | p value | HR | 95% CI | | p value | HR | 95% CI | | p value | HR | 95% CI | | p value |
| Age | 1.00 | 0.95 | 1.06 | 0.92 | | | | | 0.97 | 0.91 | 1.03 | 0.32 | | | | |
| Operated eye | 1.13 | 0.43 | 3.00 | 0.81 | | | | | 1.69 | 0.61 | 4.66 | 0.31 | | | | |
| Sex | 1.98 | 0.63 | 6.23 | 0.24 | | | | | 1.3 | 0.46 | 3.65 | 0.62 | | | | |
| Glaucoma type | 2.34 | 0.87 | 6.31 | **0.09** | 1.65 | 0.57 | 4.76 | 0.35 | 0.7 | 0.24 | 2.02 | 0.51 | | | | |
| Pre-operative BCVA | 1.88 | 0.28 | 12.53 | 0.51 | | | | | 0.74 | 0.14 | 3.96 | 0.73 | | | | |
| Pre-operative VF | 1.00 | 0.88 | 1.14 | 0.98 | | | | | 0.99 | 0.93 | 1.07 | 0.87 | | | | |
| **Pre-operative IOP** | 0.66 | 0.51 | 0.86 | **<0.01** | 0.70 | 0.52 | 0.94 | **0.02** | 0.81 | 0.65 | 1.01 | **0.06** | 0.77 | 0.61 | 0.97 | **0.02** |
| Pre-operative medication score | 0.77 | 0.51 | 1.15 | 0.19 | | | | | 0.78 | 0.55 | 1.11 | 0.17 | | | | |
| Operator | 1.99 | 1.21 | 3.28 | **0.01** | 1.44 | 0.81 | 2.54 | 0.21 | 1.64 | 0.99 | 2.71 | **0.05** | 0.45 | 0.15 | 1.37 | 0.16 |
| Transient IOP elevation | 0.69 | 0.09 | 5.49 | 0.73 | | | | | 2.06 | 0.45 | 9.34 | 0.35 | | | | |
| Coagula/Hyphema | 16.25 | 1.69 | 156.7 | **0.02** | 8.36 | 0.75 | 93.11 | 0.08 | 1.37 | 0.31 | 6.03 | 0.68 | | | | |

Cox proportional hazard model

HR: Hazard Ratio

CI: Confidence interval

Variables in the univariate analysis with *P* values ≤ 0.1 were entered into the multivariate analysis.

Higher baseline IOP is less likely to result in failure.

known to promote ocular surface disease and dry eye [21], diminishing the individual's quality of life [22]. A reduction in the number of glaucoma medications can attenuate these disadvantages of topical medication.

Previous studies have reported on iStent and iStent inject, a pre-modified model of inject W [23–26] (Table 8). Some studies suggested that iStent inject is superior in terms of IOP reduction and the percentage of patients managed without glaucoma medication [23, 24], while others reported comparable results [25, 26]. According to these studies [23, 26], both devices lowered the IOP and the number of glaucoma medications. However, no prospective randomized controlled trials have investigated this aspect, and some aspects of the superiority of inject over iStent in terms of IOP reductions and reductions in glaucoma medication are unclear. A meta-analysis of iStent alone [27] reported that two insertions were superior to one with respect to both IOP reduction and number of glaucoma medications. Hence, it is possible that the amount of aqueous humor that is bypassed is more marked with two injections than with a single iStent insertion, resulting in greater IOP reduction. However, in the present study, there was no difference in the outcomes of the iStent and inject W-treated eyes. Since the study included a large number of NTG eyes, and the pre-operative IOP was 14.8 ± 2.2 mmHg in the iStent and 15.0 ± 2.8 mmHg in the inject W groups, it is necessary to consider the possibility that the effect of both devices was limited if the pre-operative IOP was lower. The finding that a higher pre-operative IOP is less likely to cause failure is a factor influencing the surgical outcome supports this result.

Subsequently, we considered the difference between iStent and inject W in terms of technique. The iStent showed 62% insertion success on the first try, 32% on the second try, and 6% on the third try [28], while only one stent could be inserted in 9.3% of iStent inject-treated eyes [26]. Furthermore, according to previous studies on iStent inject, 88% of devices that were correctly inserted during surgery were confirmed to be in a good position by post-surgical gonioscopy [29]. These micro-stents are subject to intraoperative malpositioning, which results in

Table 8. Comparison of surgical results between iStent and inject W.

| | Follow up | N (eyes) | | Glaucoma type [iStent/inject] (%) | | IOP (mmHg) | | Medication | | |
|---|---|---|---|---|---|---|---|---|---|---|
| | | iStent | Inject (inject W) | POAG OHT | NTG | iStent | inject (inject W) | iStent | inject (inject W) | |
| Guedes RAP. Clin Opthalmol 2021 | 2 | 39 | 43 | 61.4/83.7 | 0 | 16.4 →14.8 | 17.7 →13.1 | 1.7 →0.5 | 2.2 →0.7 | IOP reduction is better with inject than with iStent |
| David M. Ophthalmol Ther 2019 | 12M | 67 | 70 | 83.6/84.3 | 1.5/1.4 | 18.4 →14.2 | 20.4 →14.4 | 1.8 →0.3 | 1.3 →0.1 | inject outperforms iStent in IOP reduction and rate of no medication |
| Hooshmand J. Clin Experiment Opthalmol 2019 | 12M | 145 | 100 | 100 | 0 | 18.9 →16.6 | 18.9 →16.9 | 1.7 →0.6 | 1.6 →0.7 | Equivalent results |
| Shalaby WS. Indian J Opthalmol 2021 | 12M | 122 | 75 | 82/84 | 9.8/9.3 | 16.2 →15.1 | 15.4 →14.0 | 2.0 →1.3 | 1.9 →1.5 | Equivalent results |
| **This study** | 12M | 55 | 105 | 72.7/61.0 | 27.3/49.0 | 14.8 →13.7 | 15.0 →13.8 | 2.5 →1.1 | 2.9 →1.7 | Equivalent results |

※ Except for this study, the results are for iStent and inject

POAG: Primary Open Angle Glaucoma; OHT: Ocular hypertension; NTG: Normal Tension Glaucoma; IOP: Intraocular pressure

failure of implantation or luminal obstruction and decreased efficacy [30]. The differences between inject W and iStent are as follows: 1) the number of insertions is one for iStent but two for inject W, 2) the number of outlet lumens is one for iStent but four for inject W, which allows the aqueous humor to flow in more directions and increases access to CCs, and 3) inject W is smaller and has direct access to CCs, and 4) iStent is placed at an angle during insertion, while inject W is inserted straight, making the procedure easier. The iStent inject W's injector is also equipped with two implants, usually utilized simultaneously, thereby reducing the impact of single implant blockage.

## Safety

No serious complications, such as choroidal detachment, suprachoroidal hemorrhage, bleb-related complications, endophthalmitis, or hypotony, were recorded compared to the incidence reported in traditional filtering surgery [9–11]. No additional surgery was required for anterior chamber hemorrhage or transient IOP elevation, which improved with observation or glaucoma medication. The BCVA also did not interfere with the effect of cataract surgery, consistent with the study by Guedes et al. [31]. The VF was also maintained. iStent and inject W showed no difference in complications between the groups, similar to the safety results described in previous studies [23–26].

## Factors affecting cumulative survival

In this study, we found that both devices were less likely to fail when the pre-operative IOP was higher, consistent with previous studies [23, 26, 32, 33]. These findings demonstrated that trabecular stent efficacy could be augmented in eyes with higher IOP, wherein the high-pressure gradient across the stent endings leads to more aqueous outflow and subsequently, greater IOP reduction. On the contrary, eyes with low baseline IOP have a lower pressure gradient,

which does not allow significant aqueous outflow through the stent. Some studies have demonstrated that older age is also associated with failure [26], but this was not a significant factor in our study. Success rates were also not influenced by the surgeon in our study, and the first-generation iStent was reported to have comparable results between senior residents and supervisors [34], suggesting that the surgical outcomes are not dependent on the surgeon's skill.

## Limitation

One of the limitations of the present study is that it incorporated a retrospective, non-randomized case series design. We also did not compare our results with cataract surgery alone. Phacoemulsification alone also reduces post-operative IOP [14, 35], but IOP reduction by phacoemulsification alone reportedly is less effective in maintaining the post-operative VF, ultimately leading to progression of VF decay [36]. As mentioned in the introduction, previous prospective studies have shown that cataract surgery with iStent or inject versus cataract surgery alone is superior in lowering the IOP and reducing the number of glaucoma medications. A previous study reported that no complications could compromise the visual improvement benefits of cataract surgery [32]. In addition, iStent and inject W are cost-effective compared to cataract surgery alone in terms of the patient's financial burden and healthcare economics [37–40]. In Japan, inject was superior to cataract surgery alone in terms of cost-effectiveness [41]. Furthermore, MIGS, including the iStent, is said to improve the patient's quality of life [42, 43]. In the future, we would like to examine the difference in cost-effectiveness and the effect of VF maintenance for iStent and inject W.

The relatively low incidence of complications compared to the overall number of patients could have obfuscated accurate assessment of the differences between the two groups. While it is evident that the procedure is relatively safe, a more precise evaluation of significant differences in complications between the two groups would require a larger cohort.

Eyes in which the iStent and inject W were not inserted correctly intraoperatively were also excluded. The learning curves of both devices and the stability of the surgical technique were also not evaluated. In iStent inject W, the interval between the two inserted devices was not standardized at 1–3 hours, depending on the case, which may also affect the results. The absence of abnormal device positioning or occlusion at each follow-up point was also not confirmed. In addition, the pre-operative IOP was lower than previously reported [23–26], which may have limited the effectiveness of the iStent and inject W and prevented accurate evaluation of the difference between the two devices. Additionally, it is worth noting that the present study did not calculate the sample size, which may have resulted in poor statistical power. To address this limitation, future studies should consider enhancing the statistical power by increasing the sample size and observing patients over an extended time period.

## Conclusions

This real-world retrospective analysis provides data on the comparative effectiveness and safety of iStent and iStent inject W combined with phacoemulsification. Over a 12-month post-operative follow-up, we found that both devices produced sustained and significant reductions in IOP and glaucoma medication while maintaining excellent safety. No statistically significant differences were observed in the outcome measures between the two groups. Lower baseline IOP was a predictor of surgical failure.

The findings provided evidence of clinically and statistically significant IOP decline, medication reduction, and excellent safety after either iStent or iStent inject W with cataract surgery. These were achieved in a real-world setting, enabling generalizability to other

ophthalmic practices. A prospective study with a larger population and longer follow-up is necessary to validate these findings.

## Supporting information

**S1 Dataset.**
(XLSX)

## Acknowledgments

We thank the participants of the study. We also would like to thank Editage ([www.editage. com](www.editage.com)) for English language editing.

## Author Contributions

**Conceptualization:** Shuu Morita.

**Data curation:** Shuu Morita, Ikari Riyu.

**Formal analysis:** Shuu Morita, Yoshihito Sakanishi.

**Methodology:** Shuu Morita, Satoshi Watanabe.

**Project administration:** Shuu Morita.

**Supervision:** Nobuyuki Ebihara.

**Writing – original draft:** Shuu Morita.

**Writing – review & editing:** Yoshihito Sakanishi.

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
