## [Decision Letter · Decision Letter 0]

5 Apr 2023

PONE-D-23-04944Comparative evaluation of iStent versus iStent inject W combined with phacoemulsification in open angle glaucomaPLOS ONE

Dear Dr. Morita,

Thank you for submitting your manuscript to PLOS ONE. After careful consideration, we feel that it has merit but does not fully meet PLOS ONE’s publication criteria as it currently stands. Therefore, we invite you to submit a revised version of the manuscript that addresses the points raised during the review process.

We look forward to receiving your revised manuscript.

Kind regards,

Aparna Rao

Academic Editor

PLOS ONE

Journal Requirements:

2. In the ethics statement in the Methods, you have specified that verbal consent was obtained. Please provide additional details regarding how this consent was documented and witnessed, and state whether this was approved by the IRB.

3. We note that Figures 1.1, 1.2, 2.1 and 2.2 in your submission contain copyrighted images. All PLOS content is published under the Creative Commons Attribution License (CC BY 4.0), which means that the manuscript, images, and Supporting Information files will be freely available online, and any third party is permitted to access, download, copy, distribute, and use these materials in any way, even commercially, with proper attribution. For more information, see our copyright guidelines: http://journals.plos.org/plosone/s/licenses-and-copyright.

a. You may seek permission from the original copyright holder of Figures 1.1, 1.2, 2.1 and 2.2 to publish the content specifically under the CC BY 4.0 license. 

Additional Editor Comments:

*Comments from PLOS Editorial Office: We note that one or more reviewers has recommended that you cite specific previously published works. As always, we recommend that you please review and evaluate the requested works to determine whether they are relevant and should be cited. It is not a requirement to cite these works. We appreciate your attention to this request.*

Reviewers' comments:

Reviewer's Responses to Questions

**Comments to the Author**

1. Is the manuscript technically sound, and do the data support the conclusions?

Reviewer #1: Yes

Reviewer #2: Partly

2. Has the statistical analysis been performed appropriately and rigorously? 

Reviewer #1: Yes

Reviewer #2: No

3. Have the authors made all data underlying the findings in their manuscript fully available?

Reviewer #1: Yes

Reviewer #2: No

4. Is the manuscript presented in an intelligible fashion and written in standard English?

Reviewer #1: Yes

Reviewer #2: No

5. Review Comments to the Author

Reviewer #1: This is an interesting article comparing the safety and efficacy of the 1st and 2nd generation iStent devices. Both devices have a ton of data supporting their use as a MIGS device but the data comparing the 1st versus 2nd generation is minimal. This data is helpful as it helps to better understand the efficacy of additional stenting and compare the design of each device. It is also timely with the introduction of the iStent infinite, a recently FDA approved 3rd generation iteration of the iStent platform that allows for implantation of 3 stents.

1. Introduction: I would remove the discussion regarding the comparison between the design of each device to make the introduction more concise. If wanting to include in the paper I would add more detail to the subheading regarding surgical technique and surgical device

2. Discussion: I think the discussion can be made considerably more concise. Instead of simply restating the results in an organized fashion, I would try and summarize efficacy into one paragraph, safety into another and explore factors for failure in the third to make it better for readers of the manuscript.

3. Results: I would consider stratifying results by baseline IOP. For good example see citation [1]. This would be helpful to understand how a patient with baseline IOP ranging from 18-20 did and see the overall breakdown of baseline IOP.

Ferguson TJ, Swan RJ, Bleeker A, Dockter Z, Karpuk KL, Schweitzer J, Ibach M, Berdahl JP. Trabecular microbypass stent implantation in pseudoexfoliative glaucoma: long-term results. Journal of Cataract & Refractive Surgery. 2020 Sep 1;46(9):1284-9.

4. Overall, I think this is a good study that offers important insight on iStent inject W outcomes. Further, this paper compared outcomes to the first-generation device and data comparing the 1st and 2nd-generation devices remain limited despite the extensive data published on each. I think with some small stylistic changes and overall cutting down on excess detail, this paper would be a meaningful and helpful addition to the literature.

Reviewer #2: As described in the title “comparative evaluation of iStent versus iStent inject W…”, the main outcome in this study is the surgical outcome between iStent versus iStent inject W. The result are no statistically significant differences between the two procedures. However, the sample size calculation does not seem to be determined before the study.

Minor poitns

In criteria of selection and patient visitation of the material and methods, the authors described “post-operative BCVA and VF were also compared to the corresponding pre-operative values on day 1, week 1, months 1, and 3, and at the final visit after surgery.” However, there are no data on each follow-up visits.

Figures 1 and 2 should be deleted from the manuscript. The readers can find the pictures by internet because they were provided by the manufacturer “Glaukos.”

The authors did not find the statistical difference of the efficacy between iStent vs iStent Inject. How many sample size do the authors estimate to detect the statistical significance? Sample size calculation is not described in the statistical analysis of the material and methods.

In Tables, legends are described. Legend sentences are removed from the tables.

In factors affecting cumulative survival in the results, preoperative IOP was a significant factor. Table 7 indicates that lower IOP is a significant factor for surgical failure. However, there are no descriptions in the result. In univariate analysis, operator is a significant factor. Who is a significant factor for surgical failure?

In Table 7, there are no units. Therefore, it is difficult to understand the values.

In discussion, although the authors describe “this study is the first report to compare the post-operative results of iStent and inject W.” However, there are several retrospective reports about iStent vs iStent inject. iStent inject W is a minor revision of iStent inject. Therefore, the efficacy of iStent inject W seems to be the same as iStent inject.

The manuscript format does not seem to be designed well. Please edit the manuscript, tables and figures again.

6. PLOS authors have the option to publish the peer review history of their article (what does this mean?). If published, this will include your full peer review and any attached files.

Reviewer #1: No

Reviewer #2: No

---

## [Author Response · Author response to Decision Letter 0]

20 Jul 2023

Response to Reviewers

Response to the Editor

Response: Thank you for your suggestion. We have ensured that the manuscript meets PLOS ONE's style requirements and have referred to the journal’s style templates.

2. In the ethics statement in the Methods, you have specified that verbal consent was obtained. Please provide additional details regarding how this consent was documented and witnessed, and state whether this was approved by the IRB.

Response: Thank you for your valuable comment. We have added the necessary IRB-approval permit number in the manuscript.

3. We note that Figures 1.1, 1.2, 2.1 and 2.2 in your submission contain copyrighted images. All PLOS content is published under the Creative Commons Attribution License (CC BY 4.0), which means that the manuscript, images, and Supporting Information files will be freely available online, and any third party is permitted to access, download, copy, distribute, and use these materials in any way, even commercially, with proper attribution. For more information, see our copyright guidelines: http://journals.plos.org/plosone/s/licenses-and-copyright.

 a. You may seek permission from the original copyright holder of Figures 1.1, 1.2, 2.1 and 2.2 to publish the content specifically under the CC BY 4.0 license. 

Response: Thank you for your suggestion. As we were unable to obtain the copyright permissions, we have removed the figures 1.1, 1.2, 2.1 and 2.2 from the revised manuscript, as per your suggestion.

Reviewer 1 comments

1. Introduction: I would remove the discussion regarding the comparison between the design of each device to make the introduction more concise. If wanting to include in the paper I would add more detail to the subheading regarding surgical technique and surgical device

Response: We have deleted the discussion regarding the comparison between the design of each device to make the introduction more concise.

2. Discussion: I think the discussion can be made considerably more concise. Instead of simply restating the results in an organized fashion, I would try and summarize efficacy into one paragraph, safety into another and explore factors for failure in the third to make it better for readers of the manuscript.

Response: We have reorganized the Discussion section to concisely describe the efficacy, safety, and factors for failure. 

3. Results: I would consider stratifying results by baseline IOP. For good example see citation [1]. This would be helpful to understand how a patient with baseline IOP ranging from 18-20 did and see the overall breakdown of baseline IOP.

Ferguson TJ, Swan RJ, Bleeker A, Dockter Z, Karpuk KL, Schweitzer J, Ibach M, Berdahl JP. Trabecular microbypass stent implantation in pseudoexfoliative glaucoma: long-term results. Journal of Cataract & Refractive Surgery. 2020 Sep 1;46(9):1284-9.

Response: We had already attempted to stratify the baseline IOP of 18–20 mmHg and study them separately, but the small number of n (eyes) to stratify prevented meaningful statistical analysis.

Reviewer 2 Comments

1. As described in the title “comparative evaluation of iStent versus iStent inject W…”, the main outcome in this study is the surgical outcome between iStent versus iStent inject W. The result are no statistically significant differences between the two procedures. However, the sample size calculation does not seem to be determined before the study.

Response: No sample size calculations were done in advance for this study. Therefore, as you have indicated, the power of this study may have been inadequate. We would like to add that as a limitation.

Minor points

1. In criteria of selection and patient visitation of the material and methods, the authors described “post-operative BCVA and VF were also compared to the corresponding pre-operative values on day 1, week 1, months 1, and 3, and at the final visit after surgery.” However, there are no data on each follow-up visits.

Response: For visual acuity, Table 4 shows the results at the time of the method description. Regarding VF, we have corrected the method, as it was not measured at the time points described.

2. Figures 1 and 2 should be deleted from the manuscript. The readers can find the pictures by internet because they were provided by the manufacturer “Glaukos.”

Response: We have removed the figures from the Manuscript. 

The authors did not find the statistical difference of the efficacy between iStent vs iStent Inject. How many sample size do the authors estimate to detect the statistical significance? Sample size calculation is not described in the statistical analysis of the material and methods.

Response: No sample size calculations were performed in advance for this study. Therefore, as you have indicated, the power may have been inadequate. We would like to add that as a limitation.

3. In Tables, legends are described. Legend sentences are removed from the tables.

Response: I would like to inform you that the tables are correctly formatted as required by the journal submission guidelines. Legends are provided below the table as required.

4. In factors affecting cumulative survival in the results, preoperative IOP was a significant factor. Table 7 indicates that lower IOP is a significant factor for surgical failure. However, there are no descriptions in the result. In univariate analysis, operator is a significant factor. Who is a significant factor for surgical failure?

Response: Although there seems to be a difference in the outcomes between surgeons, the results are based on univariate analysis. Since confounding factors may be involved, a multivariate analysis was performed, and operator was not a significant factor influencing the results. In other words, we do not believe that the difference in performance between surgeons in this study is significant.

5. In Table 7, there are no units. Therefore, it is difficult to understand the values.

Response: The table shows Hazard Ratios and Confidence intervals, and thus there are no units.

6. In discussion, although the authors describe “this study is the first report to compare the post-operative results of iStent and inject W.” However, there are several retrospective reports about iStent vs iStent inject. iStent inject W is a minor revision of iStent inject. Therefore, the efficacy of iStent inject W seems to be the same as iStent inject.

Response: We have added a sentence at the beginning of the Discussion to acknowledge the presence of several retrospective studies.

7. The manuscript format does not seem to be designed well. Please edit the manuscript, tables and figures again.

Response: I have checked the submission guidelines again and ensured full compliance with the formatting requirements.

Editor’s Comments

We've checked your submission and before we can proceed, we need you to address the following issues:

1. We notice that your manuscript file was uploaded on February 6, 2023. Please can you upload the latest version of your revised manuscript as the main article file, ensuring that does not contain any tracked changes or highlighting. This will be used in the production process if your manuscript is accepted. Please follow this link for more information: http://blogs.PLOS.org/everyone/2011/05/10/how-to-submit-your-revised-manuscript/

Response: We have uploaded the correct file.

2. Please upload a Response to Reviewers letter which should include a point by point response to each of the points made by the Editor and / or Reviewers. (This should be uploaded as a 'Response to Reviewers' file type.) Please follow this link for more information: http://blogs.PLOS.org/everyone/2011/05/10/how-to-submit-your-revised-manuscript/

Response: We have uploaded a Response to Reviewer letter containing our responses to the reviewers’ and editor’s comments. 

3. Please amend the title either on the online submission form or in your so that they are identical.

Response: We have revised the title and resubmitted the manuscript.

4. Please upload a copy of Figures 2 and 3 which you refer to in your text on pages 5 and 17. Or if the figure is no longer to be included as part of the submission please remove all reference to it within the text.

Response: We have corrected the figures for consistency between what we quote in the text and the figure uploaded.

---

## [Decision Letter · Decision Letter 1]

2 Nov 2023

PONE-D-23-04944R1Comparative evaluation of iStent versus iStent inject W combined with phacoemulsification in open angle glaucomaPLOS ONE

Dear Dr. %Morita%,

Thank you for submitting your manuscript to PLOS ONE. After careful consideration, we feel that it has merit but does not fully meet PLOS ONE’s publication criteria as it currently stands. Therefore, we invite you to submit a revised version of the manuscript that addresses the points raised during the review process.

Dear authors,

Your manuscript has completed peer-review process and comments are provided below. You will see that the reviewers though find some merits but have expressed many serious concerns. Hence, I recommend going through the critiques appended below and take appropriate action. Additionally, adequate addressing of

concerns with additional data and evidence from literature is vital for final decision. Also, consider following points while revising manuscript. Thank you

We look forward to receiving your revised manuscript.

Kind regards,

Rajiv R. Mohan, Ph.D.

Academic Editor

PLOS ONE

Additional Editor Comments:

Dear authors,

Your manuscript has completed peer-review process and comments are provided below. You will see that the reviewers though find some merits but have expressed many serious concerns specially reviewer-1. Hence, I recommend going through the critiques appended below and take appropriate action. Additionally, adequate addressing of concerns with additional data and evidence from literature is vital for final decision. Also, consider following points while revising manuscript. Thank you

Reviewers' comments:

Reviewer's Responses to Questions

**Comments to the Author**

1. If the authors have adequately addressed your comments raised in a previous round of review and you feel that this manuscript is now acceptable for publication, you may indicate that here to bypass the “Comments to the Author” section, enter your conflict of interest statement in the “Confidential to Editor” section, and submit your "Accept" recommendation.

Reviewer #2: All comments have been addressed

Reviewer #3: All comments have been addressed

2. Is the manuscript technically sound, and do the data support the conclusions?

Reviewer #2: Yes

Reviewer #3: Yes

3. Has the statistical analysis been performed appropriately and rigorously? 

Reviewer #2: No

Reviewer #3: Yes

4. Have the authors made all data underlying the findings in their manuscript fully available?

Reviewer #2: Yes

Reviewer #3: Yes

5. Is the manuscript presented in an intelligible fashion and written in standard English?

Reviewer #2: Yes

Reviewer #3: Yes

6. Review Comments to the Author

Reviewer #2: Although the reviewer commented the sample size calculation for the comparison of the efficiencies between the two types of iStent. But, the authors have not shown it. The authors do not seem to be able to conclude the difference of the efficiencies between them.

In discussion: The authors described “this study is the first report to compare the post-operative results of iStent and inject W.” But, a similar study has already been reported by other authors “Kanda S, Fujishiro T, Karakawa A, Nakagawa S, Ishii K. Clinical Outcomes of Phacoemulsification in Japanese Patients Receiving First and Revised Second-Generation Trabecular Microbypass Stents. Asia Pac J Ophthalmol (Phila). 2023 May-Jun 01;12(3):279-283. doi: 10.1097/APO.0000000000000611. Epub 2023 May 15. PMID: 37494255.”

Reviewer #3: Please see attached document, also pasted here:

1. Introduction, line 63-65: “Recently, minimally invasive glaucoma surgeries (MIGS) have emerged, and they are expected to reduce IOP more safely and effectively.”

When compared to trabeculectomies or tube shunts, MIGS carry a better safety profile, however, it is not clear that they are more effective, so you could consider deleting the word “effectively”.

2. Introduction

A suggestion that I wanted to bring up, though not critical- Is the iStent still available for use, or is only the iStent inject W sold? I realize that in a previous revisions of your manuscript, the diagram of the iStent and iStent inject W with measurements of the lumen sizes may have been removed, which is okay. The differences between the iStent and iStent inject W are discussed in the discussion section (lines 345-352). However, I believe that readers may want to know why the two types of stents are being compared in the first place, especially if they have a choice between the two types of stents. For example, is it simply whether one stent is better than two? Or is it the stent itself. Therefore, it might be helpful to consider describing the differences between the two in the introduction section. Additionally, maybe a quantification of differences in parameters hypothesized to affect pressure and flow, such as difference in lumen size in microns between the two (inject W with 80 micron central inlet and 80 micron outlet with four 50 micron side outlets and 360 micron length) could be included.

3. “Criteria of selection and patient visitation”, line 109

I would recommend changing the “visitation” wording to “patient visits” or “Inclusion and Exclusion Criteria” for example. Visitation has multiple meanings.

4. Line 116-118, “In addition, all patients who underwent cataract surgery with either iStent or inject W insertion and those who could not be inserted iStent or two inject W were excluded from the analysis.”

Wording could be clarified to say that for patients in whom the iStent or iStent inject W was attempted but unsuccessfully inserted were excluded from the study.

5. In discussion of complications- Given the number of total patients and the overall rarity of complications, it is difficult to comment whether or not there is any difference. For example, coagula in anterior chamber and “Other” (iritis, macular edema, etc.) does have a large magnitude difference between the two groups, though not statistically significant.

6. Discussion, line 293-294. Typo: “But,this However, this study is the first report to compare the post-operative results of iStent and inject W.”

“But,this” should be deleted.

7. Discussion, 295. “mid-term” results.

This should be quantified as an exact month or year, instead of mid-term (ambiguous length of time).

8. Limitations – Line 394-395.

Just a comment - the discussion that no minimum clock hour distance between the two stents for the inject W was set is important, as there is some evidence of differential and segmental flow of aqueous out of Schlemm’s Canal. This means that stents placed further apart may potentially have a higher probabilistic chance of accessing high flow regions rather than simply providing more surface area of outflow. Alternatively, maybe placing two stents side by side in an area of good flow provides even better pressure reduction, would in a large group of patients cancel out the effect of having higher probabilistic chance of accessing high flow areas.

9. Discussion, line 369-371: “On the contrary, the eyes with low baseline IOP have a lower pressure gradient, which does not allow significant aqueous outflow through the stent.”

This is a very interesting point regarding outflow and percent reduction of IOP, as people with lower IOPs may potentially be “closer” to their episceral venous pressure “floor” and thus have a reduced ability to lower pressure effectively by methods that do not bypass the EVP, as you pointed out.

I enjoyed reading the paper and I believe that it addresses the question you all sought out to answer. Thank you!

7. PLOS authors have the option to publish the peer review history of their article (what does this mean?). If published, this will include your full peer review and any attached files.

Reviewer #2: No

Reviewer #3: No

---

## [Author Response · Author response to Decision Letter 1]

2 Dec 2023

Review Comments to the Author

Reviewer #2: Although the reviewer commented the sample size calculation for the comparison of the efficiencies between the two types of iStent. But, the authors have not shown it. The authors do not seem to be able to conclude the difference of the efficiencies between them.

Answer: Thank you for your comment. We appreciate your insights. In this study, we did not conduct a pre-study sample size calculation, which may have been responsible for the lower statistical power. Therefore, we have addressed this limitation on Lines 432-4 of the manuscript.

In discussion: The authors described “this study is the first report to compare the post-operative results of iStent and inject W.” But, a similar study has already been reported by other authors “Kanda S, Fujishiro T, Karakawa A, Nakagawa S, Ishii K. Clinical Outcomes of Phacoemulsification in Japanese Patients Receiving First and Revised Second-Generation Trabecular Microbypass Stents. Asia Pac J Ophthalmol (Phila). 2023 May-Jun 01;12(3):279-283. doi: 10.1097/APO.0000000000000611. Epub 2023 May 15. PMID: 37494255.”

Answer: When we initially submitted our report, there were no similar studies published in the literature. However, as of now, one such study is available. Therefore, we have revised the statement from "this study is the first report to compare the post-operative results of iStent and inject W" to “However, our study is valuable owing to the scarcity of studies comparing the outcomes of iStent and inject W [15]."

Reviewer #3: Please see attached document, also pasted here:

1. Introduction, line 63-65: “Recently, minimally invasive glaucoma surgeries (MIGS) have emerged, and they are expected to reduce IOP more safely and effectively.”

When compared to trabeculectomies or tube shunts, MIGS carry a better safety profile, however, it is not clear that they are more effective, so you could consider deleting the word “effectively”.

Answer: Thank you for your comments. Due to the potential for misunderstanding, we will omit the term "effectively" from the sentence in question.

2. Introduction

A suggestion that I wanted to bring up, though not critical- Is the iStent still available for use, or is only the iStent inject W sold? I realize that in a previous revisions of your manuscript, the diagram of the iStent and iStent inject W with measurements of the lumen sizes may have been removed, which is okay. The differences between the iStent and iStent inject W are discussed in the discussion section (lines 345-352). However, I believe that readers may want to know why the two types of stents are being compared in the first place, especially if they have a choice between the two types of stents. For example, is it simply whether one stent is better than two? Or is it the stent itself. Therefore, it might be helpful to consider describing the differences between the two in the introduction section. Additionally, maybe a quantification of differences in parameters hypothesized to affect pressure and flow, such as difference in lumen size in microns between the two (inject W with 80 micron central inlet and 80 micron outlet with four 50 micron side outlets and 360 micron length) could be included.

Answer: Thank you for your feedback. Both iStent and iStent inject W are currently available in Japan. A previous reviewer suggested that the figures related to iStent and inject W should be removed, and that the textual discussion on structural differences is unnecessary. Consequently, these sections have been deleted from the revised version. As you pointed out, we believe that the potential for improved outcomes lies in the simple increase in outflow due to the straightforward insertion of one or two stents and refinement of the device's shape for more direct access to Schlemm's canal.

3. “Criteria of selection and patient visitation”, line 109

I would recommend changing the “visitation” wording to “patient visits” or “Inclusion and Exclusion Criteria” for example. Visitation has multiple meanings.

Answer: The subsection heading has been modified based on your suggestion. 

4. Line 116-118, “In addition, all patients who underwent cataract surgery with either iStent or inject W insertion and those who could not be inserted iStent or two inject W were excluded from the analysis.”

Wording could be clarified to say that for patients in whom the iStent or iStent inject W was attempted but unsuccessfully inserted were excluded from the study.

Answer: I have made the following revisions as suggested. 

“In addition, patients in whom the iStent or iStent inject W was attempted but unsuccessfully inserted were excluded from the study.”

5. In discussion of complications- Given the number of total patients and the overall rarity of complications, it is difficult to comment whether or not there is any difference. For example, coagula in anterior chamber and “Other” (iritis, macular edema, etc.) does have a large magnitude difference between the two groups, though not statistically significant.

Answer: Thank you for your suggestion, we have noted this point in the Limitation as follows (Lines 393-6).

“The relatively low incidence of complications compared to the overall number of patients could have obfuscated accurate assessment of the differences between the two groups. While it is evident that the procedure is relatively safe, a more precise evaluation of significant differences in complications between the two groups would require a larger cohort.”

6. Discussion, line 293-294. Typo: “But,this However, this study is the first report to compare the post-operative results of iStent and inject W.”

“But,this” should be deleted.

Answer: We have deleted the erroneous phrase. 

7. Discussion, 295. “mid-term” results.

This should be quantified as an exact month or year, instead of mid-term (ambiguous length of time).

Answer: Thank you very much for this insightful comment. We have provided specific numerical values, as requested.

8. Limitations – Line 394-395.

Just a comment - the discussion that no minimum clock hour distance between the two stents for the inject W was set is important, as there is some evidence of differential and segmental flow of aqueous out of Schlemm’s Canal. This means that stents placed further apart may potentially have a higher probabilistic chance of accessing high flow regions rather than simply providing more surface area of outflow. Alternatively, maybe placing two stents side by side in an area of good flow provides even better pressure reduction, would in a large group of patients cancel out the effect of having higher probabilistic chance of accessing high flow areas.

Answer: Thank you for this insightful comment. I believe you are correct. Currently, we are investigating whether the outcomes differ based on the distance between the placement of two iStent inject W devices in our facility. We hope to report our findings once they are compiled.

9. Discussion, line 369-371: “On the contrary, the eyes with low baseline IOP have a lower pressure gradient, which does not allow significant aqueous outflow through the stent.”

This is a very interesting point regarding outflow and percent reduction of IOP, as people with lower IOPs may potentially be “closer” to their episceral venous pressure “floor” and thus have a reduced ability to lower pressure effectively by methods that do not bypass the EVP, as you pointed out.

Answer: Thank you very much for your comments.

---

## [Editor Report · Decision Letter 2]

8 Jan 2024

Comparative evaluation of iStent versus iStent inject W combined with phacoemulsification in open angle glaucoma

PONE-D-23-04944R2

Dear Dr. % Morita%,

We’re pleased to inform you that your manuscript has been judged scientifically suitable for publication and will be formally accepted for publication once it meets all outstanding technical requirements.

Kind regards,

Rajiv R. Mohan, Ph.D.

Academic Editor

PLOS ONE

Additional Editor Comments (optional):

Dear authors,

I am happy to inform that your revisions/changes adequately addressed all concerns. The manuscript is accepted for publication. Congratulations!
---

## [Editor Report · Acceptance letter]

26 Jan 2024

PONE-D-23-04944R2 

PLOS ONE

Dear Dr. Morita, 

I'm pleased to inform you that your manuscript has been deemed suitable for publication in PLOS ONE. Congratulations! Your manuscript is now being handed over to our production team.

Kind regards, 

on behalf of

Dr. Rajiv R. Mohan 

Academic Editor

PLOS ONE